# Protective Potential of *Cicerbita alpina* Leaf Extract on Metabolic Disorders and Oxidative Stress in Model Animals

**DOI:** 10.3390/ijms251910851

**Published:** 2024-10-09

**Authors:** Dimitrina Zheleva-Dimitrova, Alexandra Petrova, Yonko Savov, Reneta Gevrenova, Vessela Balabanova, Georgi Momekov, Rumyana Simeonova

**Affiliations:** 1Department of Pharmacognosy, Faculty of Pharmacy, Medical University of Sofia, 2 Dunav St., 1000 Sofia, Bulgaria; rgevrenova@pharmfac.mu-sofia.bg (R.G.); vbalabanova@pharmfac.mu-sofia.bg (V.B.); 2Department of Pharmacology, Pharmacotherapy and Toxicology, Faculty of Pharmacy, Medical University of Sofia, 2 Dunav St., 1000 Sofia, Bulgaria; alexpetrova.work@gmail.com (A.P.); gmomekov@pharmfac.mu-sofia.bg (G.M.); 3Institute of Emergency Medicine “N. I. Pirogov”, Bul. Totleben 21, 1000 Sofia, Bulgaria; Yonko_savov@hotmail.com

**Keywords:** diabetes mellitus type 2, *Cicerbita alpina*, oxidative stress, antioxidant enzymes

## Abstract

Metabolic disorders (MDs) include disease states such as diabetes mellitus, obesity, dyslipidemia, hyperuricemia, etc., affecting about 30% of the planet’s population. The purpose of the present study was to investigate the protective potential of *Cicerbita alpina* leaf extract (ECA) against chemically induced type 2 diabetes in Wistar rats. Additionally, some biochemical parameters in the blood serum and liver, as well as histopathological investigation, were also performed. Quantitative analysis of the major compounds in the used extract was performed using ultrahigh-performance liquid chromatography-diode array detection (UHPLC-DAD) analyses using the external standard method. *C. alpina* extract revealed a beneficial effect on MDs, lowering blood sugar levels and MDA quantity in the liver, increasing the reduced glutathione level, and increasing antioxidant enzyme activity. Cichoric acid (CA) (91.93 mg/g dry extract (de) ± 4.64 mg/g de) was found to be the dominant compound in the extract, followed by caftaric (11.36 ± 2.10 mg/g de), and chlorogenic acid (CGA) (9.25 ± 0.05 mg/g de). In conclusion, *C. alpina* leaf extract (ECA) is rich in caffeoyltartaric and caffeoylquinic acids and provides beneficial effects on the diabetic animal model.

## 1. Introduction

*Cicerbita alpina* (L.) Wallr. (*Lactuca alpina* (L.) A.Gray), or alpine chicory, is a perennial herbaceous plant belonging to the subtribe Lactucinae, tribe Lactuceae of the family Asteraceae. It is distributed in the alpine zone at altitudes from 1000 to 2000 m a.s.l. [1]. In the Northeastern regions of Italy, young shoots of alpine chicory are traditionally collected in the wild for consumption as a local delicacy. Thus, the species has commercial value as a vegetable, and cultivation trials have been conducted [2]. Based on the literature survey of *C. alpina* chemical compounds, the following secondary metabolites have been isolated and identified: phenolic acids such as chlorogenic, 3,5-dicaffeylquinic, caffeyltartaric (caftaric) and mostly cichoric acid [2], sesquiterpene lactones: 8-acetyl-15-β-D-glucopyranosyllactucin and 11β,13-dihydrolactucin [3], 8-acetyl-lactucin, 8-acetyl-11β,13-dihydrolactucin and lactucin [4], and sonhuside A (in aerial parts) [5]. Additionally, the roots of alpine chicory are rich in furanocoumarins such as imperatorin, isoimperatorin, oxypeucedanin, and ostruthol [3]. Recently, a profound LC-HRMS analysis of the alpine chicory extracts from leaves and flowering heads depicted more than 100 specialized natural compounds, including acylquinic and acyltartartaric acids, flavonoids, sesquiterpene lactones (STLs) (lactucin and dihydrolactucin), their derivatives, and coumarins [6].

*Lactuca* species are characterized by their ability to produce bitter sesquiterpene lactones with various structures, including guaianolides, germacranolides, and eudesmanolides [7]. These phytochemicals, such as lactucin and other STLs, are responsible for *Lactuca’s* pain-relief, anti-inflammatory, and sedative properties, as well as their bitterness and antifeedant activity [8,9]. Additionally, the presence of phenolic compounds, like caffeic acid derivatives, in these plants indicates their potential for radical scavenging and UV radiation protection [10,11]. Also, cichoric acid, found in these species, is a natural food-derived compound, an ester of caffeic and tartaric acids. It is associated with a wide range of potential health benefits, including antioxidant, anti-inflammatory, obesity-preventative, and neuroprotective effects [12,13], as well as with prospective application for regulating glucose homeostasis, improving diabetes and its complications [14,15]. Regarding *C. alpina*, previous research identified high levels of antioxidant caffeic acid derivatives in its edible shoots. In DPPH assays, dicaffeoyl derivatives such as cynarin demonstrated greater free radical scavenging activity compared to their monocaffeoyl counterparts, caftaric, and chlorogenic acids [16]. As a part of our ongoing investigation of some Asteraceae medicinal plants, we reported the presence of cichoric acid in *Cicerbita alpina* 80% methanol leaf extract and its strong antioxidant capacity [6]. Moreover, the extract revealed a prominent inhibitory capacity towards some enzymes, involved in numerous diseases, including metabolic disorders (lipase), Alzheimer’s disease (acetyl- and butyrylcholinesterases), and hyperpigmentation (tyrosinase). Hence, the phenolic compounds could be a major component of the antioxidant potentials of foods [17], and so alpine chicory could be a natural source of antioxidants.

As a part of our in-depth investigation of Asteraceae species and *C. alpina*, herein we investigate the in vivo antidiabetic and antioxidant capacity of ECA, using a model of type 2 diabetes, induced with streptozotocin in Wistar rats.

## 2. Results and Discussion

### 2.1. Glucose Concentration in the Blood

The results of blood glucose measurement with test strips during the 28-day period of the experiment are presented in Table 1. During the 28-day period of the experiment, the blood glucose level in the control rats and in the animals treated with the two doses of extract remained unchanged. After the induction of diabetes on the seventh day, the blood sugar level began to rise gradually. In the diabetic animals during the second, third, and fourth weeks, the blood sugar was 33%, 41%, and 64% higher, respectively, compared with the blood sugar of the control animals during the corresponding week. The administration of the acarbose and the low dose of the extract did not significantly change the glucose level in the second and third weeks of the experiment compared to the diabetic animals for this time period. More significant differences in blood sugar levels have been observed during the last week of the experiment. The acarbose and the low dose of the extract reduced the glucose level by 25% and 23%, respectively, compared to the untreated DT2 rats, while the application of the extract at a dose of 500 mg/kg reduced the blood sugar by 28% compared to the diabetic animals and about 5% compared to the diabetic rats treated with acarbose.

The hypoglycemic potential of the main compounds of the ECA, cichoric acid, caftaric, and chlorogenic acid, have been described in numerous studies [15,18,19,20]. There are various mechanisms for this antidiabetic action. CA inhibits pancreas apoptosis and adjusted islet function in diabetic mice, leading to an increase in insulin generation and secretion [15]. It has also been suggested that CA increases insulin sensitivity by improving mitochondrial function [21]. CA also has a hypoglycemic effect due to a peripheral effect on muscle glucose uptake [14]. In our previous in vitro experiments, we reported that the ECA has a good alfa glucosidase-inhibiting activity [6]. Chicoric acid has been reported to have a beneficial effect on glucose transport. Chicoric acid promoted insulin-independent glucose uptake and Akt phosphorylation by post-translational regulation of AMPKα in C2C12 myotubes and improved glucose tolerance in the mice model. The dicaffeoyltartaric acid promoted AMPKα activation in L6 myocytes, and the ability of AMPKα can activate Akt, if the effect of chicoric acid (12.5 and 25 µM) on Akt was dependent upon AMPKα has been determined [13].

Caftaric acid at a concentration range from 10^−10^ to 10^−6^ M decreased high blood glucose levels, increasing insulin secretion. In addition, the acid does not increase insulin secretion in low glucose concentrations. The compound caused gene expression of insulin regulatory genes (IRS1, INSR, INS1, INS2, and PDX1), proliferative genes, and glucose transporter 2 (GLUT2) in pancreatic islets and, consequently, the acid plays a significant role in diabetes therapy [18].

Chlorogenic acid (CGA) could be considered an insulin sensitizer that potentiates insulin action similar to the pharmacological action of metformin. In contrast, CGA lowers blood glucose levels by directly inhibiting G-6-Pase activity with the associated effects of hepatic glycogenolysis and gluconeogenesis [22]. CGA can improve glucose tolerance, improving sensitivity to insulin. Impaired glucose tolerance and insulin resistance have been associated with differences in the hepatic mRNA expression of the spliced variants of the insulin receptor. CGA inhibites the *α*-glucosidase activity and intestinal glucose uptake in vitro. Furthermore, CGA is thought to stimulate the secretion of glucagonlike peptide-1 (GLP-1), which is known to have a beneficial effect on the response to glucose in pancreatic beta cells. CGA activates the AMPK and is able to strengthen the activity of carnitine palmitoyl transferase [22].

The studied *Cicerbita alpina* extract did not show significant effects in healthy animals (without STZ) because it does not change the insulin secretion and the subsequent hypoglycemia (low blood sugar levels).

Although the decrement in blood glucose in the present in vivo experiment was not dramatic, daily consumption of *C. alpina* for a prolonged period of time would contribute to improving the glycemic control in patients in a prediabetic state, as an adjunct to physical exercises or to conservative therapy with oral antidiabetic drugs such as acarbose.

### 2.2. Serum Biochemical Parameters

The administration of streptozotocin and the development of DT2 in our experiment were associated with an imbalance in all biochemical parameters examined in the serum from the experimental animals (Table 2). On day 29, after euthanasia of the animals and blood collection for biochemistry, dyslipidemia was detected, manifested by a rise in the concentration of total cholesterol and triglycerides by 52% and by 140% (*p* < 0.05), accordingly compared with control rats (Table 2). It is well known that the levels of triglyceride (TGs) and very low-density lipoprotein cholesterol (VLDL-C) were significantly elevated, and high-density lipoprotein cholesterol (HDL-C) was significantly depressed in Type 2 Diabetic patients [23]. Hepatic function was also found to be impaired with increased activity of transaminases ASAT and ALAT by 62% and 145% (*p* < 0.05), respectively, compared to the controls (Table 2). Diabetes affects all systems in the body, including the liver. Hyperglycemia affects the metabolism of lipids, carbohydrates, and proteins and can lead to non-alcoholic fatty liver disease (NAFLD), which can further progress to non-alcoholic steatohepatitis [24]. Research has shown an increase in ASAT and ALAT in 50–80% of NAFLD patients; ALT levels were higher than AST levels for NAFLD patients [25]. Disorders in kidney function are established through increased urea and creatinine levels by 125% and 24% (*p* < 0.05), respectively, compared to control serum. Defective pentose phosphate shunt in patients with uremia provoked lipid peroxidation of polyunsaturated fatty acids in the cell membranes and increased the level of MDA [23]. Oral administration of both doses of the ECA throughout the experimental period did not result in any deviation in the biochemical parameters studied in healthy animals. In rats with STZ-induced diabetes, administration of both doses of ECA improved almost to the same extent the values of the studied biochemical parameters, and no dose-dependent effect was observed. Cholesterol levels decreased by about 25% (*p* < 0.05) and triglycerides by about 55% (*p* < 0.05) in contrast to diabetic animals (Table 2). Lowering triglycerides and cholesterol is primarily associated with a reduction in the risk of major vascular events, particularly stroke and myocardial infarction. Reducing the level of triglycerides also reduces the risk of pancreatitis, as well as steatosis and subsequent more serious liver problems. Other than the antidiabetic effect, an antihyperlipidemic action has been described for both cichoric and chlorogenic acids. It is suggested that both effects of cichoric and chlorogenic acids are mediated by activation of AMP-activated protein kinase (AMPK) [26,27]. Transaminase activity declined by around 20% (*p* < 0.05) at both doses of ECA compared to diabetic controls. Xiao et al., 2013 [28] and Ding et al., 2020 [27] found that CA alleviates high-fat diet-induced liver injury by mitigating hepatic steatosis and by decreasing the activity of fatty acid synthase (FAS), ALAT and ASAT. The level of urea decreased significantly by about 14% and creatinine by about 20% compared to diabetic animals. Most probably, these effects were also due to the presence of CA, which notably lessened the levels of kidney injury markers [15].

### 2.3. Oxidative Stress Markers and Antioxidant Enzymes Activity in the Liver

Induction of DT2 with STZ was associated with prominent oxidative stress, which was manifested by significantly elevated production of MDA by 51% and a drop in the level of GSH by 38%, compared to controls (Table 3). Decreased levels of GSH in diabetic rat livers have been linked with reduced GST, GPX and GR activity and with the cumulation of advanced glycation end-products (AGEs), protein oxidation products (POPs), and lipid peroxidation (LPO) [29]. LPO causes a prominent increase in MDA and thiobarbituric acid-reactive substances (TBARS) in the livers of diabetic patients. The hyperglycemia accompanying the hyperlipidemia we observed in the present experiment could be the crucial factor for the augmented production of lipid peroxides and free radicals, i.e., MDA [30]. It was observed that in diabetic patients, the production of ROS increased and promoted lipid peroxidation. Intermolecular cross-linking of collagen by MDA results in its stabilization and permits further glycation. This, in turn, expands the potential of glycated collagen to launch further lipid peroxidation [31].

Significantly reduced activity of the antioxidant enzymes GPx and SOD by 52% and 31%, respectively, in contrast with the control animals (Table 3), was also observed in the present work. Similar findings have been observed and reported in diabetic patients [23]. Hyperglycemia in these patients can boost free radical production. Glucose itself triggers an auto-oxidative reaction and production of free radicals, resulting in superoxide anion (O_2_^−^) and hydrogen peroxide (H_2_O_2_) [32]. GPx accelerates the reduction of lipid peroxides involving GSH. In addition to catalase, H_2_O_2_ could be destroyed by GPx, leading to increased GSH utilization and its consequent depletion in cells, including in hepatocytes [33]. Using this mechanism, the exhaustion of GSH disrupts (most often reduces) the activity of antioxidant enzymes. A significant decrease of SOD activity could be a result of the progressive glycation of enzymatic proteins. About 50% of SOD in erythrocytes of diabetic patients is glycated, leading to its reduced activity [34]. Decreased SOD activity increases the level of superoxide radicals that inactivate GPx [35]. Decreased SOD and GPx activity in hyperglycemic patients increases the ROS concentration, which promotes oxidation-induced liver injury [36]. However, it is noteworthy that CAT activity is up 19% relative to untreated controls. Likidlilid et al., 2010 considered increased CAT activity as a mechanism to cope with the deleterious effect of oxidative damage [37]. The elevated CAT activity in a diabetic state is evidence of increased ROS production. A beneficial effect of treating diabetic animals with ECA was revealed by the significant increase in GSH levels by 27%, while the quantity of MDA decreased by 30% compared to diabetic rats (Table 3). The treatment of diabetic animals with a low dose (250 mg/kg) of ECA led to a significant reduction in MDA production by 25% compared to diabetic rats but did not affect GSH content. Treatment of non-diabetic animals with the two doses of ECA alone did not alter the parameters studied, nor did acarbose alter these parameters in STZ-treated rats.

Oral administration of ECA on diabetic rats demonstrated an additional antioxidant potential, proved by an increased antioxidant enzyme activity almost to the control levels. The lower dose of ECA increased significantly the GPx and SOD activity by 33% and 49%, respectively, whereas administration of the higher dose of ECA elevated both enzyme activity by 70% and 56%, compared to diabetic animals. Three weeks of administration of acarbose 5 mg/kg did not influence the antioxidant enzyme activity (Table 3).

The observed antioxidant effects of ECA are most probably due to the higher content of cichoric acid. There are numerous data on the antioxidant and organoprotective potential of cichoric acid (53.16 mg/g dry extract (de) ± 4.60 mg/g de) [15,38,39,40]. CA has a protective role in oxidative stress and inflammation via the AMPK/Nrf2/NFκB signaling pathway. Activation of AMPK could reduce mitochondrial oxidative harm and apoptosis could prevent oxidative stress-induced senescence by improving autophagic flux and NAD (+) homeostasis [27]. CA also triggers the Nrf2-Keap1 pathway and raises the expression of antioxidant enzymes [15]. It was proved that CA increased the activity of antioxidant enzymes, including NQO-1, HO-1, SOD, CAT and GSH, via the inhibition of Keap-1 and activation of Nrf2 nuclear translocation. These examinations were of relevance given that CA reversed the inflammatory processes via the activation of antioxidant response [15].

### 2.4. Histopathological Observations

The obtained results did not reveal pathological alterations in the observed sections of the liver from control and ECAhd-treated rats (Figure 1A,B). The sections of livers from diabetic animals were characterized by sinusoidal dilatation (sd) and congestion (sc) with portal inflammation (pi) and small droplets focal steatosis (fs) (toxic type, provoked by STZ application) (Figure 1C). Less pronounced local steatosis (ls) in the livers of acarbose-treated diabetic rats was noticed (Figure 1D). Sinusoidal dilatation and portal inflammation were not seen in this group. Livers from diabetic rats treated with high dose ECA (Figure 1E) were with almost restored cellular architecture.

In the histological sections from the pancreas of the control animals and the rats treated only with the high dose of the extract, no pathological abnormalities were observed. The complete islets (iL) were uniformly arranged with substantial pancreatic β-cells (Figure 1A). In rats with chemically induced metabolic alterations, pancreatic tissue atrophy, which appeared with a decreased count of β-cells, substituted by fatty tissue (ft) was noted (Figure 1C). Regained pancreatic morphology in the acarbose-treated animals was detected, but it still had dissolved and deformed cells (dc) (Figure 1D). Succeeding the higher dose (500 mg/kg) ECA application, the islets’ morbid structures were effectively restored with an increased number of pancreatic β-cells (Figure 1E). These tissue-protective effects of ECA were probably also due to the presence of cichoric acid. Zhu et al., discovered that CA can reverse pancreatic cell apoptosis and improve insulin secretion [15].

### 2.5. UHPLC-DAD Study

An UHPLC-DAD for the quantitative analysis of the *C. alpina* extract (ECA) main compounds was carried out. Generally, thirteen secondary metabolites, including five acyltartaric acids (ATAs) (1, 3, 4, 5, and 6), six acylquinic acids (AQAs) (2, 7, 9, 10, 11, and 12), and two flavones luteolin (13), and its O-glycoside (8) were identified in ECA. Based on the UV spectra and previous LC-HRMS data, compounds 3, 4, and 5 were categorized as ATAs, while 7 and 12 were categorized as AQAs [6,41]. A chromatogram at three different wavelengths (360 nm, 310 nm, and 280 nm) of ECA is depicted in Figure 2. The content of the assayed compounds is revealed in Table 4. Cichoric acid (6) was the major compound in ECA, followed by caftaric acid (1), chlorogenic acid (2), and 1,5-dicaffeoylquinic acid (10).

## 3. Materials and Methods

### 3.1. Plant Material and Sample Extraction

Plant material was collected at Vitosha Mt., Bulgaria, and identified as previously described [6,36]. Then, 50 g of air-dried powdered leaves were extracted twice for 15 min with 80% MeOH by ultra-sound (1:20 *w*/*v*, 100 kHz, Biobase UC-20C, Jinan, Shandong, China). The obtained extract was concentrated, defatted with CH_2_Cl_2_, and lyophilized (lBiobase BK-FD10P, Jinan, Shandong, China). The obtained crude extract was 6.5 g. The lyophilized extract of *C. alpina* (ECA) was then used for UHPLC-DAD analyses and in vivo tests.

### 3.2. Chemicals

Streptozotocin, acarbose, bovine serum albumin (fraction V), beta–Nicotinamide adenine dinucleotide 2′-phosphate reduced tetrasodium salt (NADPH), reduced glutathione (GSH), oxidized glutathione (GSSG), glutathione reductase (GR), acarbose, and cumene hydroperoxide were obtained from Sigma Chemical Co. (Taufkirchen, Germany). 2,2-Dinitro-5,5 dithiodibenzoic acid (DTNB) and methanol were acquired from Merck (Darmstadt, Germany). α-Glucosidase was assured by Sigma-Aldrich (St. Louis, MO, USA). Acetonitrile, formic acid, and methanol were taken from Chromasolv (Sofia, Bulgaria). The reference standards used for the identification of chlorogenic, caftaric, cichoric, 1,5-, 3,4-, and 3,5-dicaffeoylquinic acids were bought from Phytolab (Vestenbergsgreuth, Germany). Luteolin 7-*O*-glucoside was supplied from Extrasynthese (Genay, France). All reagents were of analytical grade.

### 3.3. Animals

Forty male Wistar rats (180–200 g) were supplied by the National Breeding Center, Sofia, Bulgaria, and used for the present experiment. Rats were housed in cages on a 12/12 light/dark cycle under standard laboratory conditions (ambient temperature 20 ± 2 °C and humidity 72 ± 4%). Standard rat chow and drinking water were provided in sufficient quantities. All research was conducted in accordance with the principles set out in the European Convention for the Protection of Vertebrate Animals Used for Experimental and Other Scientific Purposes (ETS 123) and was authorized by the Bulgarian Food Safety Agency (Permit No. 346 of 28.02.2023).

### 3.4. Induction of Diabetes Type 2

This variant of the disease was provoked on the seventh day after the beginning of the experiment using the sequential intraperitoneal (i.p.) application of 110 mg/kg of nicotinamide (NA) (b.w.) followed by 45 mg/kg b.w. streptozotocin (STZ) [42]. Rats with 7 mmol/L or higher blood glucose levels were regarded as diabetic.

### 3.5. Experimental Design

Our preliminary studies found that a single oral gavage of ECA to five animals at a dose of 5000 mg/kg did not lead to mortality or toxic effects. For this reason, for the four-week treatment of the rats, we used doses that represented 1/20 and 1/10 of 5000 mg/kg, or 250 and 500 mg/kg, accordingly.

The purpose of this study was to investigate the possible protective effects of two doses of ECA administered alone to male normal rats and rats with STZ-induced diabetes type 2 (DMT2). The intestinal glucosidase inhibitor acarbose was used as a positive control. Thirty-five rats were divided into seven groups of five animals (n = 5) each as follows:

Group 1—controls, with free access to drinking water and standard rat chow;

Group 2—rats orally gavaged with 250 mg/kg ECA (ECAld) for 4 weeks:

Group 3—rats given a high dose of the extract (ECAhd, or 500 mg/kg) for 4 weeks.

Group 4—DMT2 pathological control group;

Group 5—DMT2 group treated orally once a day with acarbose (5 mg/kg) [43] (from 2nd to 4th week);

Group 6—diabetic rats treated with ECAld for 28 days;

Group 7—DMT2 rats treated with ECAhd for 28 days;

Blood glucose level was measured weekly with test strips.

On day 29, all experimental animals were put under anesthesia with ketamine/xylazine and euthanized. Blood for the assessment of serum biochemical parameters was centrifugated at 3000× *g* for 10 min. The separated serum was assessed using commercial kits for an automated biochemical analyzer (BS-120, Mindray, Shenzhen, China) according to the manufacturer’s instructions. Livers were removed for the evaluation of oxidative stress and antioxidant enzyme activity in the studied groups. Pieces of the livers and pancreas were also taken for histopathological analysis.

### 3.6. Serum Biochemical Investigation and Oxidative Stress Markers

Liver oxidative stress was evaluated by assessing the quantity of malondialdehyde (MDA) as described by Polizio and Peña [44]. The GSH level was evaluated using the method of Bump et al., 1983 [45]. Glutathione peroxidase (GPx) activity was assessed using the method of Tappel [46]. Catalase activity (CAT) was assessed using the method of Aebi, 1974 [47].

Superoxide dismutase activity (SOD) was evaluated according to the method of Misra and Fridovich, 1972 [48].

### 3.7. Histopathological Exploration

Pathomorphological research of livers and pancreas was done using the method described by Bancroft and Gamble (2002) [49]. The sections were observed under a high-power microscope, and photomicrographs were taken using “Olympus” CX31 and Camera “Olympus x Optical zoom” with an objective “PlanaC” 4/0.10 (Karl Zeiss, Jena, Germany).

### 3.8. UHPLC-DAD Analysis

The UHPLC-DAD analyses were performed on a Thermo Scientific (Waltham, MA, USA) Dionex UltiMate 3000 analytical system equipped with a Dionex UltiMate 3000 RS Pump (LPG-3400RS), Dionex UltiMate 3000 RS Autosampler (WPS-3000TRS), Dionex UltiMate 3000 RS Column Compartment (TCC3000RS) and Dionex UltiMate 3000 Diode Array Detector (DAD-3000). The separation, quantitative analysis, and analytical parameters were formerly reported by Mihaylova et al., 2024 [41].

Based on the closed UV spectra, the quantities of 1, 3, 4, and 5 were determined based on the calibration curve of cichoric acid (6), and 7, 10, 11, and 12 were quantified as 3,4-dicaffeoylquinic acid (9). The analytical characteristics for cichoric acid are as follows: y = 0.0966x − 1.1913; r^2^ = 0.9916.

### 3.9. Statistical Analysis

Comparisons within two groups were made using the Student’s *t*-test. One-way analysis of variance (ANOVA) with post hoc multiple group comparisons (Dunnet *t*-test) was used to assess statistical differences. Values of *p* < 0.05, *p* < 0.01, and *p* < 0.001 were considered statistically significant. Means, standard deviations, and correlations for the UHPLC-DAD were calculated using MS EXCEL 97-2003.

## 4. Conclusions

The global prevalence of metabolic diseases has risen over the past two decades, demanding new approaches and strategies that are not only effective but also locally acceptable and feasible. The present study revealed an in vivo protective potential of *Cicerbita alpina* leaf extract against diabetes mellitus type 2 in model animals. ECA demonstrated a helpful effect on DMT2, lowering blood sugar levels and MDA quantity in the liver and increased reduced glutathione level and antioxidant enzymes activity. The ECA, rich in caffeoyltartaric and caffeoylquinic acids, provided a beneficial effect on the serum biochemical parameters (cholesterol, triglycerides, ASAT, and ALAT). Our findings are evidence of the health-promoting effects of edible Cichoriae species. Consumption of cichoric, caftaric, and chlorogenic acids, in particular, could be considered a practical approach to managing the diverse elements of metabolic disorders and for their applications as additives in alternative or conventional therapy.

## Figures and Tables

**Figure 1 ijms-25-10851-f001:**
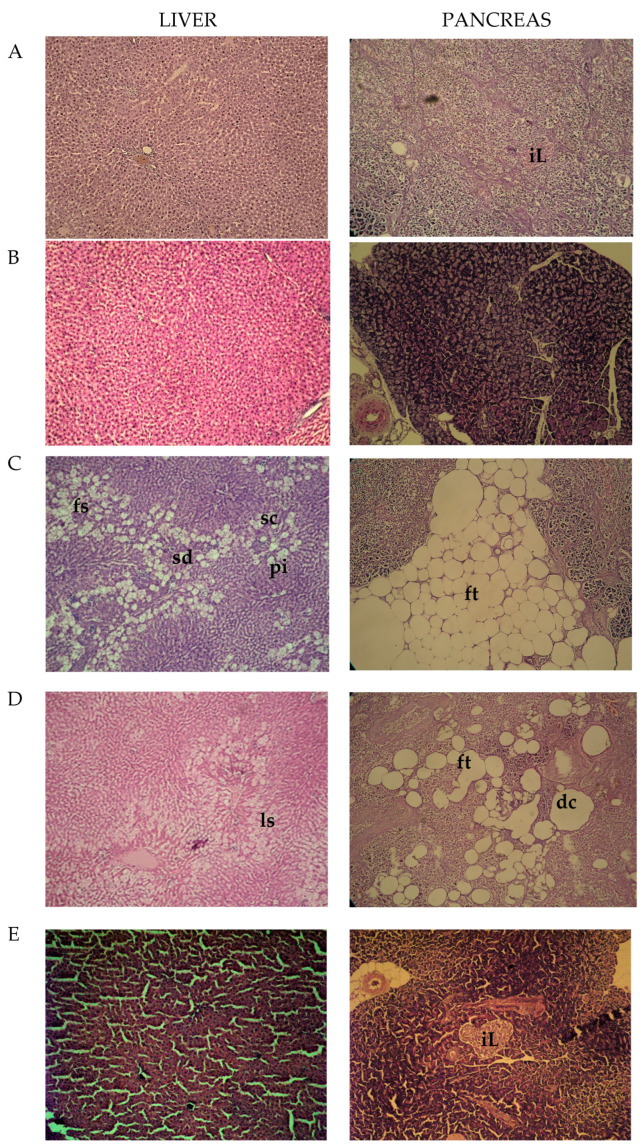
Histopathological photomicrographs of the liver and pancreas; (**A**)—control animals; (**B**)—rats treated with ECAhd: (**C**)—rats challenged with NA-STZ: (**D**)—diabetic rats, receiving acarbose; (**E**)—diabetic rats, treated with ECAhd for 28 days.

**Figure 2 ijms-25-10851-f002:**
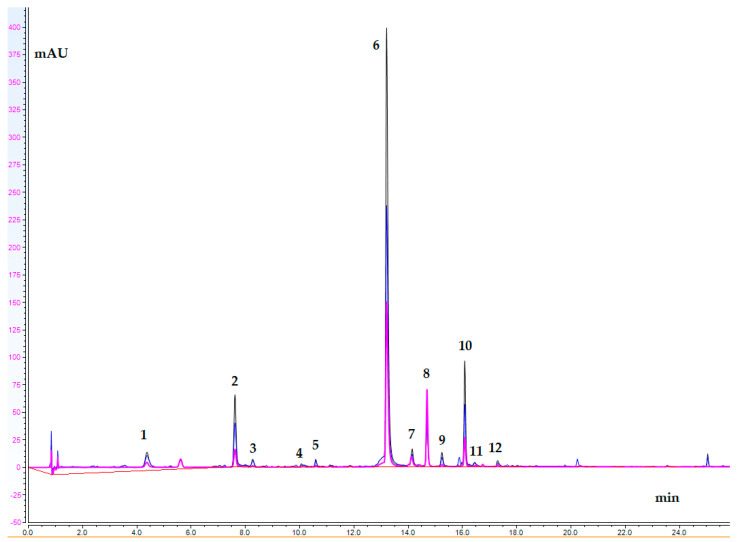
UHPLC-DAD chromatogram of ECA; wavelengths: 360 nm, 310 nm, and 280 nm (peak assignments are listed in Table 4).

**Table 1 ijms-25-10851-t001:** Changes in the blood glucose levels of the experimental animals, presented by week in mmol/L.

Groups	Week 1	Week 2	Week 3	Week 4
Control	5.9 ± 0.35	5.8 ± 0.30	5.8 ± 0.31	5.6 ± 0.19
ECAld	5.9 ± 0.23	5.7 ± 0.24	6.1 ± 0.11	6.0 ± 0.55
ECAhd	5.7 ± 0.29	5.8 ± 0.32	6.0 ± 0.17	6.1 ± 0.34
DMT2	6.0 ± 0.21	7.7 ± 0.52 **	8.2 ± 0.42 ***	9.2 ± 0.32 ***
DMT2 + Acarb	5.8 ± 0.24	7.4 ± 0.29 *	7.8 ± 0.31 ***	6.9 ± 0.22 **^+++^
DMT2 + ECAld	5.8 ± 0.13	7.2 ± 0.18 **	8.1 ± 0.16 ***	7.1 ± 0.14 ***^+++^
DMT2 + ECAhd	6.0 ± 0.29	7.2 ± 0.11 *	7.6 ± 0.23 ***	6.6 ± 0.15 **^+++#^

* *p* ≤ 0.05 vs. control; ** *p* ≤ 0.01 vs. control; *** *p* ≤ 0.001 vs. control; ^+++^
*p* ≤ 0.001 vs. DMT2; ^#^
*p* ≤ 0.05 vs. acarbose. Results are expressed as mean ± SD (n = 5). Comparisons within two groups were made using the Student’s *t*-test. One-way analysis of variance (ANOVA) with post hoc multiple group comparisons (Dunnet *t*-test) was used to assess statistical differences. Values of *p* < 0.05, *p* < 0.01, and *p* < 0.001 were considered statistically significant. Abbreviations: ECAld, extract of *C. alpina*, low dose (250 mg/kg); ECAhd, extract of *C. alpina* high dose (500 mg/kg); DMT2, diabetes mellitus type 2; DMT2 + Acarb, diabetic animals, treated with acarbose; DMT2 + ECAld, diabetic animals treated with 250 mg/kg ECA; DMT2 + ECAhd, diabetic animals treated with 500 mg/kg ECA.

**Table 2 ijms-25-10851-t002:** Changes in blood biochemical parameters.

Parameter	Controls	ECAld	ECAhd	DMT2	DMT2 + Acarb	DMT2 + ECAld	DMT2 + ECAhd
Cholesterol mmol/L	1.51 ± 0.06	1.52 ± 0.08	1.53 ± 0.07	2.29 ± 0.07 ***	1.85 ± 0.061 ***^+++^	1.77 ± 0.10 **^++^	1.66 ± 0.14 *^++#^
Triglycerides mmol/L	0.54 ± 0.03	0.52 ± 0.03	0.50 ± 0.08	1.30 ± 0.06 ***	0.60 ± 0.05 ***^+++^	0.62 ± 0.04 *^+++^	0.59 ± 0.04 ^+++^
ASAT U/L	88.8 ± 5.7	90.5 ± 3.1	87.4 ± 3.85	143.8 ± 17.8 ***	139.8 ± 16.9 ***	121.2 ± 11.5 **^+^	111.0 ± 6.32 ***^+#^
ALAT U/L	65.6 ± 6.5	66.2 ± 10.6	69.6 ± 4.8	160.6 ± 9.99 ***	147.6 ± 12.7 ***	128.4 ± 13.7 **^++#^	133.2 ± 17.1 **^++^
Urea mmol/L	6.31 ± 0.65	6.67 ± 0.87	6.61 ± 0.81	14.2 ± 0.75 ***	12.0 ± 0.32 ***^+^	12.6 ± 0.60 ***^+^	12.3 ± 0.9 ***^+^
Creatinine µmol/L	52.9 ± 2.23	50.2 ± 1.16	51.9 ± 1.68	65.5 ± 3.23 **	51.1 ± 2.4 ^++^	50.1 ± 1.5 ^+++^	52.2 ± 2.25 ^++^

* *p* ≤ 0.05 vs. control; ** *p* ≤ 0.01 vs. control; *** *p* ≤ 0.001 vs. control; ^+^
*p* ≤ 0.05 vs. DMT2; ^++^
*p* ≤ 0.01 vs. DMT2; ^+++^
*p* ≤ 0.001 vs. DMT2; ^#^
*p* ≤ 0.05 vs. acarbose. Comparisons within two groups were made using the Student’s *t*-test. One-way analysis of variance (ANOVA) with post hoc multiple group comparisons (Dunnet *t*-test) was used to assess statistical differences. Values of *p* < 0.05, *p* < 0.01, and *p* < 0.001 were considered statistically significant.

**Table 3 ijms-25-10851-t003:** Effects of ECA on the hepatic oxidative stress markers and on the antioxidant enzyme activity.

Parameter	Controls	ECAld	ECAhd	DMT2	DMT2 + Acarb	DMT2 + ECAld	DMT2 + ECAhd
MDA ^1^	3.41 ± 0.29	3.59 ± 0.32	3.51 ± 0.27	5.15 ± 0.23 ***	4.68 ± 0.42 **	3.82 ± 0.27 **^++#^	3.62 ± 0.2 *^++##^
GSH ^1^	6.64 ± 0.55	7.0 ± 0.14	7.26 ± 0.24	4.11 ± 0.3 **	4.38 ± 0.24 **	4.65 ± 0.35 *	5.24 ± 0.13 *^++##^
GPx ^2^	2.92 ± 0.28	2.86 ±0.29	2.89 ± 0.18	1.39 ± 0.15 ***	1.53 ± 0.19 **	1.85 ± 0.11 **^++#^	2.36 ± 0.17 *^++##^
CAT ^2^	5.70 ± 0.14	5.75 ± 0.29	5.5 ± 0.29	6.8 ± 0.27 **	6.66 ± 0.41 **	7.31 ± 0.13 ***^+#^	7.31 ± 0.12 *^,^***^+#^
SOD ^2^	1.66 ± 0.1	1.68 ± 0.07	1.73 ± 0.06	1.15 ± 0.11 **	1.35 ± 0.14 **	1.71 ± 0.12 ^++##^	1.79 ± 0.19 ^++#^

^1^ nmol/g tissue; ^2^ nmol/min/mg protein; * *p* ≤ 0.05 vs. control; ** *p* ≤ 0.01 vs. control; *** *p* ≤ 0.001 vs. control; ^+^
*p* ≤ 0.05 vs. DMT2; ^++^
*p* ≤ 0.01 vs. DMT2; ^#^
*p* ≤ 0.05 vs. acarbose; ^##^
*p* ≤ 0.01 vs. acarbose. Comparisons within two groups were made using the Student’s *t*-test. One-way analysis of variance (ANOVA) with post hoc multiple group comparisons (Dunnet *t*-test) was used to assess statistical differences. Values of *p* < 0.05, *p* < 0.01, and *p* < 0.001 were considered statistically significant.

**Table 4 ijms-25-10851-t004:** Content (mg/g dry extract) of compounds assayed in ECA.

No	Analyte	t_R_	Content (mg/g de)
1.	Caftaric acid	4.35	11.364 ± 2.09
2.	Chlorogenic acid	7.60	9.248 ± 0.050
3.	ATA I	8.26	2.998 ± 0.336
4.	ATA II	10.06	1.549 ± 0.086
5.	ATA III	10.59	2.538 ± 0.047
6.	Cichoric acid	13.21	91.930 ± 4.642
7.	AQA I	14.15	5.032 ± 0.116
8.	Luteolin 7-*O*-glucoside	14.70	3.006 ± 0.068
9.	3,4-diCQA	15.25	2.627 ± 0.045
10.	1,5-diCQA	16.09	9.455 ± 0.434
11.	3,5-diCQA	16.47	0.370 ± 0.005
12.	AQA II	17.30	0.909 ± 0.038

## Data Availability

The original contributions presented in the study are included in the article, further inquiries can be directed to the corresponding author/s.

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
