# Peer review of "Protective Potential of Cicerbita alpina Leaf Extract on Metabolic Disorders and Oxidative Stress in Model Animals"

_ijms, 2024, doi:10.3390/ijms251910851_

Round 1

Reviewer 1 Report

Comments and Suggestions for Authors

The manuscript "Protective Potential of Cicerbita alpina leaves extract on metabolic disorders and oxidative stress in model animal" present not novel but important information on a medicinal plant from a region from Bulgaria. Make my attention that the diabetes type 2 disease considered in the study used to alleviated two different doses, 250 and 500 mg/ kg. considered high, but they mentioned without toxic effects. The modest blood glucose diminution results obtained, should be consider for a prolonged period of time, i.e., 2 months or more. Please explain.

In relation to blood chemical parameters, it is observed modest diminished changes considering the high doses of the extract. What the authors argue about this?

In relation to hepatic oxidative stress markers and antioxidant enzymes, it looks the same, very few changes when added the two doses of the extract. Please argue this with more convincing words.

No comments on histopathological observations.

Author Response

Response to Reviewer 1 comments

The manuscript "Protective Potential of Cicerbita alpina leaves extract on metabolic disorders and oxidative stress in model animal" present not novel but important information on a medicinal plant from a region from Bulgaria. Make my attention that the diabetes type 2 disease considered in the study used to alleviated two different doses, 250 and 500 mg/ kg. considered high, but they mentioned without toxic effects.

Response: Dear Reviewer, thanks for your valuable comments. Our so-called "high dose" 500 mg is purely mathematically higher than the other dose we used, 250 mg. In the preliminary test for acute toxicity and determination of LD50, we used a dose of 5000 mg/kg, which is the maximum dose used in toxicology. If, upon its single application to an experimental animal, there is no mortality or serious toxic effects, the investigated compound or extract is declared to be practically non-toxic. But this is for a single application. Doses that are fractions of the LD50, 1/5, 1/10, 1/20, etc., are used for repeated administration, which we have done and no toxic effects were observed in the 28-day administration.

The modest blood glucose diminution results obtained, should be consider for a prolonged period of time, i.e., 2 months or more. Please explain.

Response: Dear Reviewer, thanks for the comments. Our preliminary studies found that a single oral gavage of ECA to five animals at a dose of 5000 mg/kg did not lead to mortality or toxic effects. For this reason, for the four-week treatment of the rats, we used doses that represented 1/20 and 1/10 of 5000 mg/kg, or 250 and 500 mg/kg, accordingly. It is a common approach for the evaluation of the plant extracts anti-diabetic potential, the administration of the extracts to be orally for 28-30 days (Jaiswal et al., 2017 https://doi.org/10.1016/j.jtcme.2016.11.007; Singh  and Singh, 2010; DOI: 10.4103/0253-7613.70238) (Singh et al., 2021 https://doi.org/10.1016/j.compbiomed.2021.104462). Previouus data revealed that the majority of such experiments are conducted over 2 to 4 weeks  (https://doi.org/10.4103%2F0253-7613.40484; https://doi.org/10.1016/j.jep.2023.116385; https://doi.org/10.1093/jn/135.10.2299; https://doi.org/10.3390/antiox12010029;  https://doi.org/10.1016/j.biopha.2022.113578; https://doi.org/10.1016/j.biopha.2023.114689). Therefore, we used the presented doses for a period of 28 days.

We agree that this type of chronic disease experiment should be of longer duration. But this also means the use of a larger number of animals, because in the course of the longer experiment, some of the animals die for various reasons. This would impair the statistical processing of the results. If the dead animals were replaced with new ones, it would increase their numbers. In Europe there are restrictions on the use of experimental animals and therefore each experiment is authorized by the Regulatory Authorities. If 30 rats are allowed, 40 cannot be used, which limits our options for a long experimental period.

In relation to blood chemical parameters, it is observed modest diminished changes considering the high doses of the extract. What the authors argue about this?

As mentioned above, "high dose" is just mathematically higher. In addition to the bioactive compounds contained in it, there are also other components that are not pharmacologically active. Administered as a mixture, combination of compounds or extract, they may not be sufficient in quantity and activity to exert a more potent pharmacological effect, i.e. this "high dose" is insufficient to realize a sharper and more significant lowering of blood sugar. It is known that nutritional supplements of plant origin used for some chronic diseases cannot cause an immediate effect or have an immediate impact on the patient's health. It is known that treatments with plant extracts, teas, infusions, etc. in folk medicine requires a long period of time or are used to maintain the condition after certain biochemical parameters have been corrected with conventional drugs.

In relation to hepatic oxidative stress markers and antioxidant enzymes, it looks the same, very few changes when added the two doses of the extract. Please argue this with more convincing words.

Response: Thanks for the valuable comments. In rats with STZ-induced diabetes, administration of both doses of ECA improved almost to the same extent, the values of the studied biochemical parameters and hepatic oxidative stress markers and antioxidant enzymes, and no dose-dependent effect was observed. Generally, plant extracts are complex systems and rich source of a variety of secondary metabolites from different classes. Possible synergistic, additive or/and antagonistic effects between different bioactive compounds could be performed in the living organisms.

Additionally, these effects were previously observed in a study including 495 patients treated with acarbose, 25, 50, 100 or 200 mg t.i.d., according to the European recommendations. Surprisingly, doses of 50 mg t.i.d. were found to be equally as effective as 100 mg t.i.d. However, 200 mg t.i.d. had the strongest effect on blood glucose. Interestingly, there was a plateau of blood glucose level at a dosage of 50–100 mg t.i.d. Similarly, dosages of 50 and 100 mg t.i.d. were equally effective with regard to HbA1C control. (Fischer et al., Acta Diabetol (1998) 35: 34–40; DOI: 10.1007/s005920050098).

Other study revealed that the increasing doses of acarbose to 50 or 100 mg had no significant additional ameliorating effects on postprandial hyperglycemia. Postprandial insulin or triglyceride levels were not significantly altered with single dose acarbose treatment (Mooradian et al., 2000; Am J Med Sci;319(5):334-7. doi: 10.1097/00000441-200005000-00011.).

Therefore, the modest diminished changes considering the high and low doses of the extract on the tested parameters could be related to possible entry into the plateau of the dose-effect curve or/and interactions between different bioactive compounds. Future investigations with pure isolated compounds in more different doses are needed for the clarification of these facts.

No comments on histopathological observations.

Reviewer 2 Report

Comments and Suggestions for Authors

The manuscript “Protective Potential of Cicerbita alpina Leaves Extract on Metabolic Disorders and Oxidative Stress in Model Animals” reports on the potential of bioactive compounds from Cicerbita alpina extract in type 2 diabetes. However, the authors need to address the following issues:

  1. Why did the administration of acarbose and the low dose of the extract show no significant differences in blood glucose levels during the first few weeks?
  2. What could be the reason that the high dose of the extract showed a greater reduction in glucose levels compared to the low dose and acarbose treatment?
  3. What other mechanisms, besides those suggested in the text, could be involved in the hypoglycemic effect of cichoric, caftaric, and chlorogenic acids?
  4. Are there notable differences in the effectiveness of the extract compared to other commonly used oral antidiabetics, such as metformin?
  5. How does the 28-day administration period of the extract compare to studies that use longer durations?
  6. What is the clinical relevance of the observed reduction in triglycerides and cholesterol after extract administration?
  7. Should other biomarkers of liver and kidney function be investigated to further assess the safety of the extract in diabetes treatment?
  8. What other antioxidant measures, besides GPx and SOD activity, could be important for evaluating the extract's potential in reducing oxidative stress?
  9. Why did the extract show no significant effects in healthy animals while having an impact on diabetic animals?
  10. What are the implications of the observed results on oxidative stress markers in terms of preventing liver damage in diabetic patients?
  11. How can the increase in CAT activity in diabetic animals treated with ECA be explained, and what is its role in neutralizing reactive oxygen species?
  12. What are the expected effects of prolonged use of the extract's components (cichoric and chlorogenic acids) in humans with type 2 diabetes, considering the results observed in animals?

Author Response

Response to Reviewer 2 comments

The manuscript “Protective Potential of Cicerbita alpina Leaves Extract on Metabolic Disorders and Oxidative Stress in Model Animals” reports on the potential of bioactive compounds from Cicerbita alpina extract in type 2 diabetes. However, the authors need to address the following issues:

  1. Why did the administration of acarbose and the low dose of the extract show no significant differences in blood glucose levels during the first few weeks?

Response: Dear reviewer, thanks for the valuable comments. STZ is a broad-spectrum antibiotic that is toxic to the insulin producing β cells of pancreatic islets. It is currently used clinically for the treatment of metastatic islet cell carcinoma of the pancreas. Male rats tend to be more susceptible to STZ-induced diabetes than the females. This difference can be significant, with little or no response in females and severe hyperglycemia present in male rats receiving identical doses. Changes in rodent diet can have a large influence on sensitivity to STZ, and have been used to create type 2 diabetes models when combined with STZ doses that do not cause diabetes in normal chow-fed rats. A single moderately sized dose of STZ has been used to induce a slowly progressive diabetes mellitus. Ito et al. reported that male ICR mice receiving a 100 mg/kg injection of STZ experienced a progressive hyperglycemia with normal non-fasting serum insulin levels and preserved β cell mass in the pancreas, indicating a non-insulin dependent diabetes mellitus (NIDDM) due to an increase of insulin resistance (Ito M, Kondo Y, Nakatani A, Naruse A. New model of progressive non-insulin-dependent diabetes mellitus in mice induced by streptozotocin. Biological & pharmaceutical bulletin. 1999 Sep;22(9):988–9). We assume that during the first 1-2 weeks, the effect of intraperitoneal administration of streptozotocin was more pronounced, compared to the hypoglycemic effect of acarbose and the low dose extract. With the administration of the higher dose of ECA, other mechanisms of hypoglycemic action are probably involved that we have not studied.

  1. What could be the reason that the high dose of the extract showed a greater reduction in glucose levels compared to the low dose and acarbose treatment?

Response: Some researchers reported that inhibition of α-glucosidase by plant extracts is higher than acarbose. (Azemi ME, Khodayar MJ, Ayatamiri FN, Tahmasebi L, Abdollahi E.. 2015. Inhibitory activities of Phoenix dactylifera, Capparis spinosa, Quercus brantii, and Falcaria vulgaris hydroalcoholic extracts on α- amylase and α-glucosidase. Int J Curr Res Chem Pharm Sci. 2:19–25.).

In our previous experiment (https://doi.org/10.3390/plants12051009) we also discovered a better α-amylase and α-glucosidase inhibitory potential of C. alpina extract than acarbose, but we did not perform a kinetic study to investigate the type of inhibition of the enzymes and dose dependence. We may only speculate that the investigated ECA has dose-dependent inhibitory potential, because of which the higher dose had a more pronounced hypoglycemic effect in vivo. 

  1. What other mechanisms, besides those suggested in the text, could be involved in the hypoglycemic effect of cichoric, caftaric, and chlorogenic acids?

Response: Thanks for the valuable comments. The other mechanisms involved in hypoglycemic effect of cichoric, caftaric, and chlorogenic acids are added in the text (See Results and discussion).

The following sentences were added in the text:

“Chicoric acid has been reported to have a beneficial effect on glucose transport. Chicoric acid promoted insulin-independent glucose uptake and Akt phosphorylation by post-translational regulation of AMPKα in C2C12 myotubes, and improved glucose tolerance in the mice model. The dicaffeoyltartaric acid promoted AMPKα activation in L6 myocytes, and the ability of AMPKα can activate Akt, if the effect of chicoric acid (12.5 and 25 µM) on Akt was dependent upon AMPKα has been determined [13].

Caftaric acid at concentration range from 10-10 to 10-6 M decreased high blood glucose levels, increasing the insulin secretion. In addition, the acid does not increase insulin secretion in low glucose concentration. The compound caused gene expression of insulin regulatory genes (IRS1, INSR, INS1, INS2 and PDX1), proliferative genes and glucose transporter 2 (GLUT2) in pancreatic islets and consequently the acid plays a significant role in diabetes therapy [18].

CGA can improve glucose tolerance, improving sensitivity to insulin. Impaired glucose tolerance and insulin resistance have been associated with differences in the hepatic mRNA expression of the spliced variants of the insulin receptor. CGA inhibited the ?-glucosidase activity, intestinal glucose uptake in vitro. Furthermore, CGA is thought to stimulate the secretion of glucagon-like peptide-1 (GLP-1), which is known to have a beneficial effect on the response to glucose in pancreatic beta cells. CGA activated the AMPK, and is able to strengthen the activity of carnitine palmitoyl transferase [22]”.

  1. Are there notable differences in the effectiveness of the extract compared to other commonly used oral antidiabetics, such as metformin?

Response: Thanks for the question. The metformin's mechanism of action is the alteration of the energy metabolism of the cell. Metformin exerts its prevailing, glucose-lowering effect by inhibiting hepatic gluconeogenesis and opposing the action of glucagon. This is a possible mechanism for ECA, but we did not perform such kind of experiments, at least we did not use metformin as a positive control, to be able to claim that the same mechanism produces the hypoglycemic effect of ECA as metformin did.

It is well-known that chronic dietary treatment with acarbose, an alpha-glucosidase inhibitor, improves glucose homeostasis in the streptozotocin (STZ)-induced diabetic rat. The intestinal glucosidase inhibitor, acarbose decreases postprandial glycemia by delaying carbohydrate absorption (Wright and al., 1998; DOI: 10.1016/s0031-9384(98)00013-4). STZ causes pancreatic β-cell death (apoptosis and necrosis) via different mechanisms resulting in fasting permanent hyperglycemia, relative hypoinsulinemia (i.e., serum insulin was comparable with fasted normal animals but low related to coexisting hyperglycemia), polyuria, glycosuria, and marked (>90 %) decrease in pancreatic insulin content (Junod et al., 1969). In our previous in vitro experiments, we reported a good alfa glucosidase inhibiting activity of the ECA. Therefore, we used the glucosidase inhibitor acarbose as positive control in our investigation. On the other hand, metformin exerts its prevailing, glucose-lowering effect by inhibiting hepatic gluconeogenesis and opposing the action of glucagon (Pernicova and Korbonits, 2014; Nature Reviews Endocrinology volume 10, pages143–156; DOI: 10.1038/nrendo.2013.256). 

Previously was found that CGA decreased body weight and improved glucose tolerance and insulin resistance, and these effects were similar to those of metformin (Yan et al., 2022; Front. Endocrinol., 17 November 2022; https://doi.org/10.3389/fendo.2022.1042044) (See row 112).

  1. How does the 28-day administration period of the extract compare to studies that use longer durations?

Response: You are right, such studies cannot be extrapolated. But in our case we are comparing similar results obtained from extracts with similar phytochemical composition. We can compare the mechanisms of action of the different extracts, and since we have only conducted an in vitro test for alpha-glucosidase inhibition, we have no claims for another mechanism of action of our extract. Investigations have been done for antidiabetic action and with a shorter duration: https://doi.org/10.4103%2F0253-7613.40484; https://doi.org/10.1016/s0378-8741(03)00122-3

  1. What is the clinical relevance of the observed reduction in triglycerides and cholesterol after extract administration?

Response:  Thank you for the nice question. Lowering triglycerides and cholesterol is primarily associated with a reduction in the risk of major vascular events, particularly stroke and myocardial infarction. Reducing the level of triglycerides also reduces the risk of pancreatitis, as well as steatosis and subsequent more serious liver problems.

  1. Should other biomarkers of liver and kidney function be investigated to further assess the safety of the extract in diabetes treatment?

Response: Diabetes complications are known to be associated with micro- and macroangiopathies, including nephropathies. Damage to kidney function, in addition to an increased level of nitrogen bodies, is also characterized by albuminuria. Therefore, examination of the level of total protein, albumin, and the presence of albumin in the urine can be markers of significant impairment of both renal and hepatic synthetic function. In addition, in severe forms of diabetes, the level of acetone and other ketones in the blood increases, i.e. the presence of ketones in the urine can also be a predictor of a complication of the condition. These waste products can alter the activity of drug-metabolizing enzyme systems (CYPs) in the liver (induction or inhibition), leading to drug interactions or compromising antidiabetic treatment. Of course, such experiments with evaluation of the activity of the enzymes of the first and second phase of drug metabolism can also be done both in vitro and in vivo conditions. Naturally, evaluating the level of bilirubin, GGT and other biochemical parameters will give us a complete picture of the secondary complications of diabetes. All these parameters could be investigated to assess the toxicity or safety of the studied extract under chronic long-term exposure.

  1. What other antioxidant measures, besides GPx and SOD activity, could be important for evaluating the extract's potential in reducing oxidative stress?

Response: Oxidative stress is attributed to the imbalance between ROS levels and the availability and activity of antioxidants or antioxidant enzymes. The imbalance is induced by increased generation of free radicals or decreased antioxidant activity. The biomarkers such as Malondialdehyde (MDA), hydrogen peroxide (H2O2), isoprostanes (IsoPs) and hydroxyl radicals (OH.) evolved as potential predictors for prognosis or response to treat diseases associated with the severity. The major antioxidant enzymes, including catalases (CATs), SODs, peroxiredoxins (PRDXs), and GPXs, work cooperatively to protect cells from an excess of ROS derived from endogenous metabolism or external hazardous microenvironment. Endothelial nitric oxide synthase (eNOS), NADPH oxidase, phospholipase A2 (PLA2) play a role in the development of oxidative complications and could be measured. Selenium, copper, zinc, and manganese are essential since they act as cofactors for antioxidant enzymes. In addition, the levels of Vits.A and E, and total oxidant status (TOS) also could be measured to assess extract's potential in reducing oxidative stress.

  1. Why did the extract show no significant effects in healthy animals while having an impact on diabetic animals?

Response: Thanks for the interesting question. STZ causes β-cell toxicity, resulting in insulin deficiency (Tesch and Allen, 2007, Nephrology (Carlton) ;12(3):261-6; DOI: 10.1111/j.1440-1797.2007.00796.x). STZ causes pancreatic β-cell death (apoptosis and necrosis) via different mechanisms, including DNA alkylation, depletion of cellular NAD+ levels and thus energy deprivation, increasing oxidative stress, and increasing nitric oxide production (Ghasemi and Jeddi., 2014, Acta Physiol Hung;101(4):408-20; DOI: https://doi.org/10.17179/excli2022-5720). When diabetogenic doses of STZ (45-65 mg/kg) is injected to male white Wistar rats, a triphasic response is observed in blood glucose concentration (Junod et al., 1969; J Clin Invest;48(11):2129-39. doi: 10.1172/JCI106180.): (1) early transient hyperglycemia (2-4 h after STZ injection) probably due to adrenaline response and sudden breakdown of liver glycogen without a parallel increase in serum insulin; (2) transient hypoglycemia (7-10 h after STZ injection) due to increased serum insulin because of insulin release from necrotizing β-cells but without a decrease in pancreatic insulin content; (3) stable hyperglycemia (24 h after STZ injection and onwards); in this phase, frank diabetes characterized by fasting permanent hyperglycemia, relative hypoinsulinemia (i.e., serum insulin was comparable with fasted normal animals but low related to coexisting hyperglycemia), polyuria, glycosuria, and marked (>90 %) decrease in pancreatic insulin content.

The studied Cicerbita alpina extract did not show significant effects in healthy animals (without STZ) because it does not change the insulin secretion and the subsequent hypoglycemia (low blood sugar levels) (See Results and Discussion). Moreover, our previous investigations revealed that C. alpina contained high levels of antioxidant caffeic acid derivatives in its edible shoots. In addition, a dose of 5000 mg/kg did not lead to mortality or toxic effects (See Experimental design).

  1. What are the implications of the observed results on oxidative stress markers in terms of preventing liver damage in diabetic patients?

Response: The liver is the main detoxification organ of the body and plays an important role in controlling normal glucose homeostasis. Reduced levels of GSH in diabetic rat livers have been associated with decreased GST, GPX and glutathione reductase activity and with the accumulation of oxidative stress products, such as advanced glycation end-products (AGEs), protein oxidation products (POPs) and lipid peroxidation (LPO). A decrease in these activities within a hyperglycaemic state leads to an increase in ROS, which eventually contributes to oxidation-induced liver damage. Prevention of GSH depletion and preservation of the activity of antioxidant enzymes through the application of plant extracts rich in phenolic compounds is a prerequisite for reducing the risk of oxidative damage to the liver in diabetics.

  1. How can the increase in CAT activity in diabetic animals treated with ECA be explained, and what is its role in neutralizing reactive oxygen species?

Response: It has been demonstrated that hydrogen peroxide acts as an oxidant and damages the β cell interrupting the signaling pathway of insulin production. According to a study from Prof. Kassab's laboratory, a four-fold increase in the concentration of H2O2 was observed in type 2 diabetes mellitus patients than in the healthy controls (Msolly A. M., Kassab A. S. Hydrogen peroxide: an oxidant stress indicator in type 2 diabetes mellitus. Journal of Cardiovascular Disease. 2013;1(2):48–52.). The increase in CAT activity in diabetic rats in our experiment may be a compensatory response to increased endogenous H2O2 production in the diabetic liver, as insulin deficiency promotes β-oxidation of fatty acids with subsequent H2O2 formation.

Therefore, the high catalase activity may contribute to the pathogenesis of a particular form of pancreatic diabetes indirectly by maintaining the H2O2 concentration which would induce the synthesis of proinflammatory cytokines resulting in this specific diabetes. Furthermore, the increased catalase activity promoted by ECA administration may be an adaptive response to overcome the increased oxidative stress in the liver and pancreatic tissues.

Catalase is one of the most important antioxidant enzymes and key enzyme which uses H2O2, as its substrate. This enzyme is responsible for neutralization through decomposition of H2O2, thereby maintaining an optimum level of the molecule in the cell which is also essential for cellular signaling processes. Catalase deficiency or malfunctioning is associated with many diseases such as diabetes mellitus, cardiovascular diseases, hypertension, anemia, Alzheimer's disease, bipolar disorder, schizophrenia etc. Hydrogen peroxide catabolism protects the cells from oxidative assault, for example, by securing the pancreatic β cells from hydrogen peroxide injury.

  1. What are the expected effects of prolonged use of the extract's components (cichoric and chlorogenic acids) in humans with type 2 diabetes, considering the results observed in animals?

Response: Thanks for the question. Based on the results obtained we expect beneficial effects in the glucose levels, blood biochemical parameters (cholesterol, triglycerides, ASAT, ALAT, etc) and hepatic oxidative stress markers on the human with diabetes type 2. The administration of these bioactive compounds in combination with classical antidiabetic agents would lead to potentiation of the effects and reduction of the dose of the antidiabetic drug. In addition, the application (alone or in combination) of cichoric, caftaric, and chlorogenic acids would present beneficial effect in various metabolic disorders.

Round 2

Reviewer 2 Report

Comments and Suggestions for Authors

The authors of the manuscript “Protective Potential of Cicerbita alpina Leaves Extract on Metabolic Disorders and Oxidative Stress in Model Animals” have made the suggested corrections. The work shows improvement and can be published on the journal’s platform.